# PeerJ

# The variability of inner ear orientation in saurischian dinosaurs: testing the use of semicircular canals as a reference system for comparative anatomy

Jesús Marugán-Lobón[1,2], Luis M. Chiappe[2] and Andrew A. Farke[3]

[1] Unidad de Paleontología, Dpto. Biología, Universidad Autónoma de Madrid, Cantoblanco (Madrid), Spain
[2] The Dinosaur Institute, Natural History Museum of Los Angeles County, Los Angeles, CA, USA
[3] Raymond M. Alf Museum of Paleontology, Claremont, CA, USA

Corresponding author
Luis M. Chiappe, chiappe@nhm.org

## ABSTRACT

The vestibular system of the inner ear houses three semicircular canals—oriented on three nearly-orthogonal planes—that respond to angular acceleration stimuli. In recent years, the orientation of the lateral semicircular canal (LSC) has been regularly used to determine skull orientations for comparative purposes in studies of non-avian dinosaurs. Such orientations have been inferred based on fixing the LSC to a common set of coordinates (parallel to the Earth's horizon), given that the orientation to gravity of this sensory system is assumed constant among taxa. Under this assumption, the LSC is used as a baseline (a reference system) both to estimate how the animals held their heads and to describe craniofacial variation among dinosaurs. However, the available data in living birds (extant saurischian dinosaurs) suggests that the orientation of the LSC in non-avian saurischian dinosaurs could have been very variable and taxon-specific. If such were the case, using the LSC as a comparative reference system would cause inappropriate visual perceptions of craniofacial organization, leading to significant descriptive inconsistencies among taxa. Here, we used Procrustes methods (Geometric Morphometrics), a suite of analytical tools that compares morphology on the basis of shared landmark homology, to show that the variability of LSC relative to skull landmarks is large (ca. 50°) and likely unpredictable, thus making it an inconsistent reference system for comparing and describing the skulls of saurischian (sauropodomorph and theropod) dinosaurs. In light of our results, the lateral semicircular canal is an inconsistent baseline for comparative studies of craniofacial morphology in dinosaurs.

Subjects Evolutionary Studies, Paleontology, Zoology, Statistics
Keywords Saurischia, Dinosaurs, Skull, Geometric morphometrics, Inner ear, Anatomy, Reference system

## INTRODUCTION

Anatomical reference systems determining the spatial context by which different parts of an animal's body should be compared (horizontal and vertical axes) are essential for

comparative anatomical and phylogenetic studies. In anthropology and paleoanthropology, multiple reference systems have been devised to that end, such as Broca's plane (the plane of the optic nerves) and the Frankfurt plane (the orbito-meatal plane) (*Gould, 1981*). Anatomical studies in primates also use Reid's baseline, a standard reference system in anthropometry that is particularly common in conventional radiography and computer tomography (CT) (*Strait & Ross, 1999*). Historically, all these morphological reference systems have been established relative to a stereotyped head posture (when at rest or in alert) and Earth's gravity, yet not surprisingly, none of them is entirely congruent to the others. When orthodontists use one reference framework, be it Reid's plane or the Frankfurt plane, the system is fortunately relatively stable because anatomical variation at an intra-specific (human) scale is small. However, important inconsistencies arise when extrapolating the use of these anatomical reference systems at larger taxonomic scales.

*Lebedkin (1924)* was the first to suggest that the lateral semicircular canal (LSC) within the inner ear could serve as a proxy to estimate at rest or alert head postures, and *de Beer (1947)* argued that angular deviations between the LSC and the Earth's horizon in such a stereotyped posture are small enough to support such assumption. This notwithstanding, de Beer also found important inconsistencies for this rule, since the method could not be applied to humans, in which the deviation of the orientation of the LSC and the Earth's horizon at such stereotype postures is nearly 37° (Fig. 1). Subsequently, striking differences in the position of the labyrinths in the skulls of different species of birds led *van der Klaauw (1948)* to point out the possibility of a functional relationship between the position of the labyrinth and a stereotyped head posture. To test this hypothesis, *Duijm (1951)* undertook an inter-specific study through direct observations on living birds, and like de Beer previously, demonstrated that the craniofacial anatomy was best compared when skulls were oriented according to the animals' alert posture (*Marugán-Lobón & Buscalioni, 2006*). Nonetheless, this study also documented inconsistencies between the orientation of the LSC and that of the skull in such a stereotyped posture—the LSC showed a broad rotational spectrum when the skulls were oriented in the alert posture (Figs. 1B and 2). This notwithstanding, the author endorsed the use of a horizontal placement of the LSC as an anatomical proxy for the way in which birds hold their heads in such postures. Just as de Beer had reasoned previously, Duijm advocated that the angular deviation of the LSC from a horizontal position (approximately −19° to 30° relative to the horizon while the skull is in the alert position) was smaller than that of any other skull structure and that the mean orientation was close to 0° (Fig. 2).

De Beer and Duijm's seminal works were inspiring, and have been widely followed not just in the context of interpretations of how extinct animals held their heads but also as a reference system for descriptive morphology and anatomical comparisons (e.g., *Rogers, 1998*; *Sampson & Witmer, 2007*; *Sereno et al., 2007*; *Evans, 2006*; *Witmer & Ridgely, 2009*; *Witmer et al., 2008*). For instance, using the orientation of the LSC *Sereno et al. (2007)* estimated the 'alert' head posture of the rebbachisaurid *Nigersaurus*, discussing that this animal's head was bizarre because its face was oriented vertical to the ground. However, the estimation of head posture from the LSC is very imprecise (*Duijm, 1951*) and if this

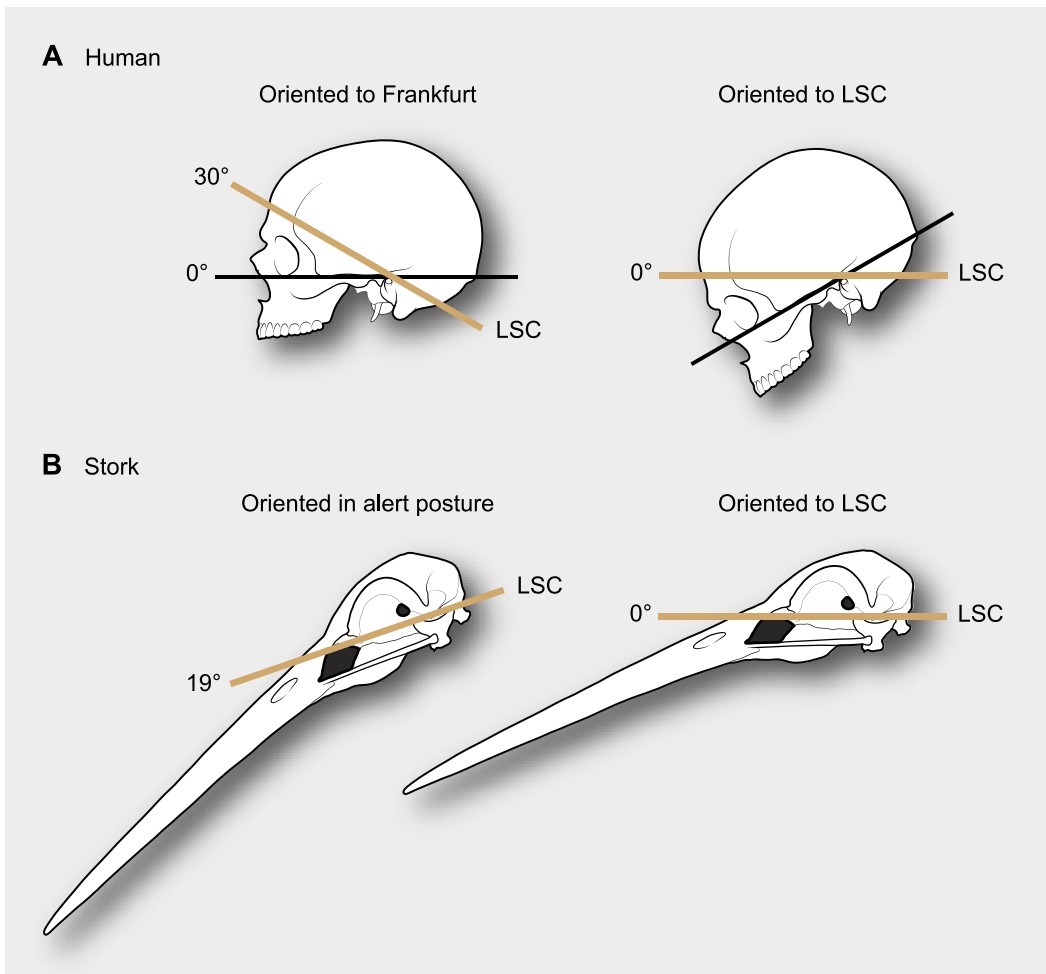

**Figure 1 Differences in reference systems in skulls.** (A) In the human skull there is a 30° difference between the Frankfurt plane and that of the LSC, thus yielding substantially different head orientations (from *de Beer, 1947*). (B) When a stork is in alert its LSC is oriented 19° above the horizon, thus when putting the LSC at 0° (horizontal) head posture differs from its alert posture (from *Duijm, 1951*).

is used as a reference system, it can lead to descriptive inconsistencies, as *Taylor, Wedel & Naish (2009)* rightfully argued. Take the human head as an example; using the LSC as a reference system (Fig. 1) one would describe the anatomy of the human skull bent forwards and downwards, as if the person looked down to the ground at a point one or two meters away (*Girard, 1923*), yet clearly, the human face is not anatomically sloped downwards. Implicitly, all these comparisons entail that the LSC is oriented differently relative to the components of the human skull, and that such may likely be the case of the sauropod *Nigersaurus*. Thus, if the orientation of the LSC is potentially unpredictable and problematic as an anatomical reference system, is there an alternative approach that can allow anatomical comparisons of the skulls to be standardized? Here we argue that methods of mathematical shape analysis are suitable for that purpose, in particular the Procrustes methods of geometric morphometrics.

Procrustes methods are widely applied in comparative morphology (*Adams, Slice & Rohlf, 2004*) and more recently, they have been used as a tool to improve the comparative study of animal behavior by filtering out uninformative body postures (*Fureix et al., 2011*). In the field of morphological research, Procrustes methods are part of the field of Geometric Morphometrics (GM), and their effectiveness relies on the comparison of configurations of biologically definable—anatomically homologous—Cartesian coordinates of points (aka landmarks) involving mathematical operations rather than concepts rooted in biological intuition or classical morphology, such as the use of recognizable postures and fixed comparative baselines (*Zelditch, Swiderski & Sheets, 2012*). To this end, GM compares configurations of 2D and 3D biologically homologous landmarks within a common reference coordinate system (the statistically computed mean configuration; *Chapman, 1990*; *Rohlf & Slice, 1990*; *Bookstein, 1991*; *Adams, Slice & Rohlf, 2004*). Such a procedure is accomplished by a least-squares estimation of translation, rotation (posture), and scaling parameters that help to optimally superimpose the landmark configurations without altering their original topology.

Here, using GM and a case study in saurischian dinosaurs, we assess whether there is any discrepancy between the orientation of skulls according to the LSC and that based on a system of coordinates provided by craniofacial landmark homology. In light of our findings, we argue that landmark homology provides a more consistent and easier method for depicting and comparatively studying anatomical systems than do classic reference systems such as the orientation of the LSC.

## MATERIALS AND METHODS

When comparing the position of the labyrinth in the skulls of different birds, one finds striking differences, and *Duijm*'s (*1951*) experiment aimed at testing if these differences in the orientation of the LSC are related to differences in head posture (*van der Klaauw, 1948*). The experiment involved three steps: (1) determining a stereotyped head posture in live birds, (2) measuring and describing skull morphologies in the stereotyped posture, and finally (3) testing if such measurements allowed estimation of the stereotyped posture from the orientation of the LSC (i.e., if there is a relationship between LSC orientation and head posture). The first step involved measuring the orientation of the head in several species of birds at the Amsterdam Zoo and in the field. These measurements were performed with binoculars equipped with a graduated arc in the lenses, a plumb-rule (used to test verticality), and a hairline to indicate the horizon. The baseline in the animal's head was the ventral edge of the beak. The second step was to replicate the observed alert head posture of the studied species ($n = 32$) in the lab, and it was achieved by reorienting the beaks of their respective skulls to the corresponding degrees measured in the stereotyped alert posture, relative to the horizon. Then, the skulls were dissected through the midline, and the orientations of the cranial floor, the clivus (i.e., the basioccipital bone), the foramen magnum, and the LSC, were all measured and analyzed. The third step focused on discussing the observed variability in orientation of the LSC in the stereotyped alert posture, which was less random (i.e., apparently more normally distributed) than that of

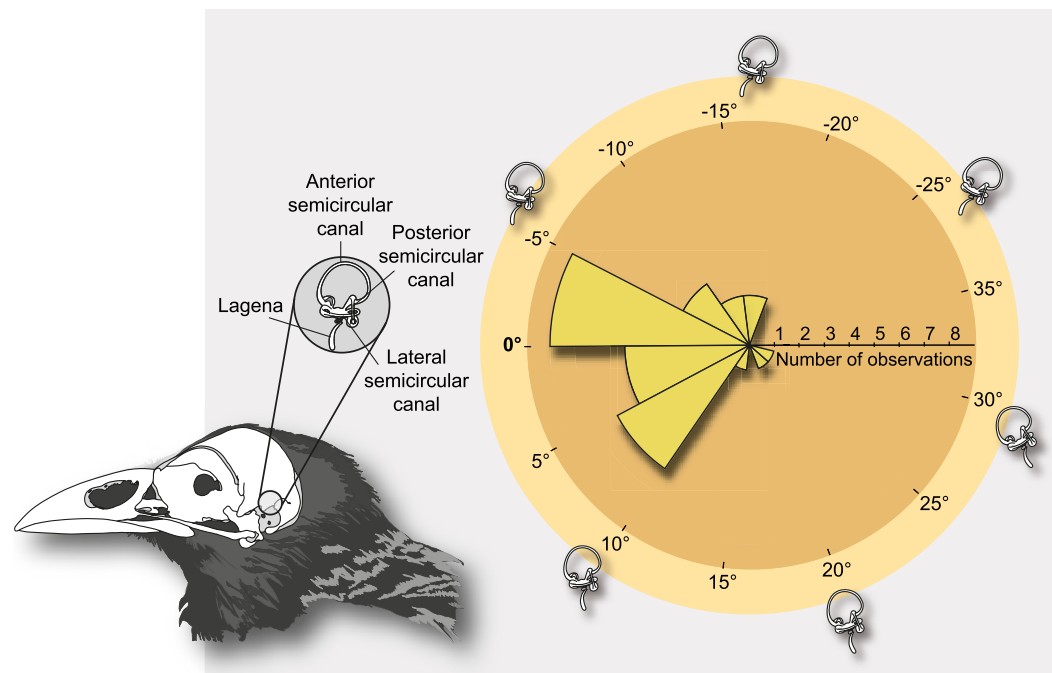

**Figure 2  Schematic depiction of the semicircular canals and polar histogram of LSC orientations in extant birds, measured in alert posture.** The semicircular canals are three interconnected tubes that define three nearly-orthogonal planes, and are part of the bony labyrinth of the inner ear. The measurements were obtained relative to the Earth's horizon by *Duijm (1951)* when the birds were in a stereotyped alert posture at a zoological garden (*n* = 29). Notice that the orientation of the lagena and the cochlear duct in the vestibular apparatus normally varies among species; in our scheme they are steady for simplicity. Although the average orientation was close to zero, LSC angles when birds are in alert approximately ranges from −19° to 30° relative to the horizon.

the other variables, yet highly variable between extremes (ca. 50°). In the original paper, all of these data are provided as skull sketches of each species with labeled lines denoting the angular orientation of the skull structures in the specific alert posture (*Duijm, 1951*; Fig. 3, p. 208). We recovered the angular data and generated Fig. 1B (the stork's head posture in alert), and Fig. 2 (polar histogram). To this end, we scanned that figure with a high-resolution flatbed scanner (at 300ppp) and measured the schemes using the digital protractor utility of TPSdig2 (2.16; *Rohlf, 2010*).

We studied a sample of saurischian dinosaurs that embraces a broad range of skull shape disparity (*Marcus, Hingst-Zaher & Zaher, 2000*) and thus, of potential semicircular canal orientations (*Duijm, 1951*). Digital pictures corresponding to 16 taxa of saurischian dinosaurs and one extant crocodile in lateral view were studied in two dimensions (Table 1). CT-scan information on the orientation of the LSC was taken from the literature for only 10 of these skulls (Table 1). Additionally, the orientation of the LSC for the stork, while in alert posture, was taken from Duijm's data. On each specimen in the sample, we digitized the coordinates of 5 homologous landmarks (Fig. 3A; see caption for their anatomical description). The landmark configurations homogeneously cover the entire skull (facial skeleton and cranium) to guarantee that additional landmarks will not lead

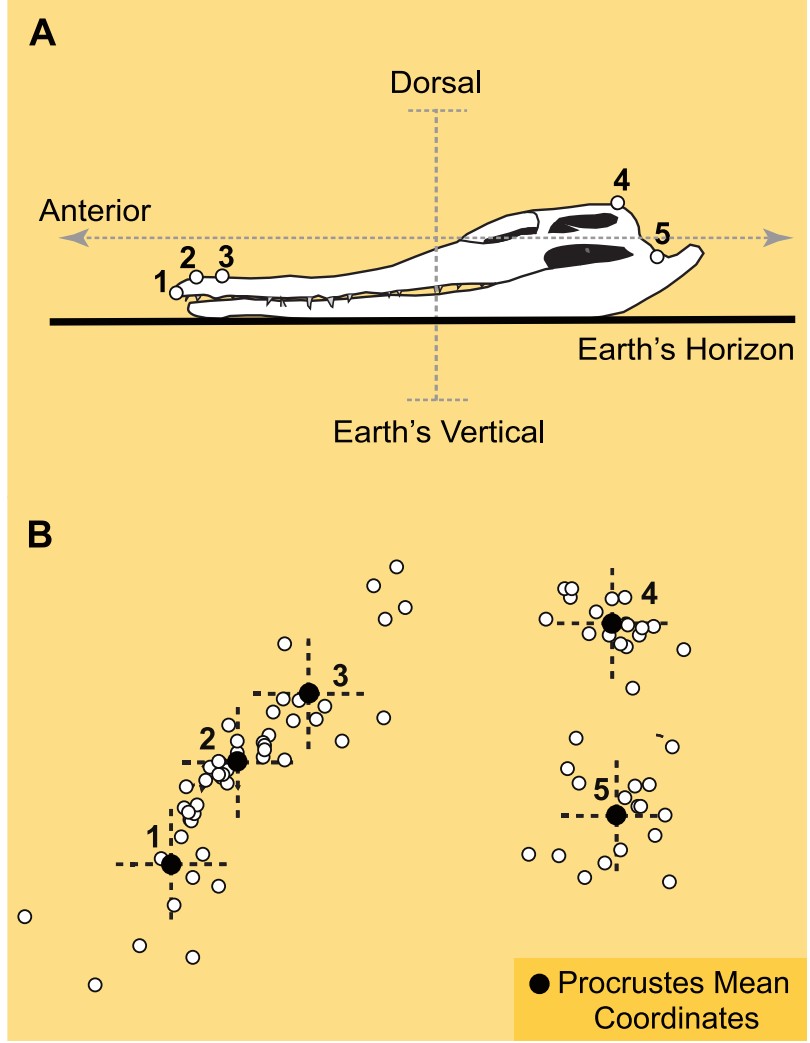

**Figure 3 Landmarks and Procrustes alignment of dinosaur skulls.** (A) Example of the configuration of $p = 5$ landmarks in lateral view of the skull of a crocodile (*C. johnstoni*), as it leans on its mandibles over a flat surface. Landmarks are: 1-tip of premaxilla, 2-margin of nasal opening closer to tip of premaxilla, 3-margin of nasal opening further from tip of premaxilla, 4-junction between supraoccipital and parietal at cranial roof, 5-mandibular articulation of quadrate. In this crocodile the orientation of the LSC relative to the horizon is ∼0° (*Witmer et al., 2008*). (B) Superimposed configurations of landmarks using Geometric Morphometrics (Generalized Procrustes methods, or GPA). The enlarged black landmarks correspond to the Procrustes mean (the consensus reference system). The *x–y* crosses at each landmark are depicted to illustrate the concomitant correspondence with the spatial directions determined by the morphological plan of the crocodile relative to the Earth's axes.

to significant differences in the results (*Marugán-Lobón & Buscalioni, 2004*). Landmarks were selected to be visible across a variety of taxa with disparate cranial anatomy. Ideally, the landmarks should be coplanar in 2D to avoid the effect of foreshortening, and our choice of coordinates is significantly close to this requirement, as it is on most studies of skull geometry that use GM. The only exception is landmark 4, but the variation of this coordinate should not alter the results since it is anatomically medial and sagittal, and

**Table 1 Studied specimens, collection numbers, and LSC orientations derived from the Generalized Procrustes Analysis.**

|    | Specimen | Collection # | LSC |
|----|----------|--------------|-----|
| 1  | *Crocodylus johnstoni* | OUVC 10425 | 0° |
| 2  | *Plateosaurus longiceps* | MB.R.1937 | — |
| 3  | *Camarasaurus lentus* | CM 11338 | −4° |
| 4  | *Nigersaurus taqueti* | MNN GAD512 | −51.1° |
| 5  | *Giraffatitan brancaii* | HMB t1 (S-II) | — |
| 6  | *Diplodocus longus* | CM 11161 | −12.5° |
| 7  | *Coelophysis bauri* | AMNH 480 | — |
| 8  | *Allosaurus fragilis* | UMNH VP 18050 | 2.3° |
| 9  | *Majungasaurus crenatissimus* | FMNH PR2100 | 0.2° |
| 10 | *Nannotyranus lancensis* | CMNH 7541 | −22.3 |
| 11 | *Tyrannosaurus rex* | AMNH 5117 | −1.1° |
| 12 | *Citpati osmolskae* | IGM 100/978 | — |
| 13 | *Incisivosaurus gauthieri* | IVPP V 13326 | 4.6 |
| 14 | *Velociraptor mongoliensis* | AMNH FR6516 | — |
| 15 | *Gallus sp.* | ZMB 77 | — |
| 16 | *Bubo virginianus* | OUVC 10220 | −9.3° |
| 17 | *Ciconia ciconia* | ZMB 253 | 3.0° |

**Notes.**

AMNH, American Museum of Natural History; HMB, MBR and ZMB, Humboldt Museum für Naturkunde; CM, Carnegie Museum; IVPP, Institute of Vertebrate Paleontology and Paleoanthropology; MNN, Musée National du Niger; FMNH, Field Museum of Natural History; OUVC, Ohio University Vertebrate Collections; IGM, Mongolian Institute of Geology; CMNH, Cleveland Museum of Natural History.

therefore restricted to vary only in a single 2D plane, comparable to that of the rest of landmarks (i.e., it will not change coronally).

The landmark coordinates were aligned by a Generalized Procrustes Analysis (*Gower, 1975*) using the program Morpheus et al. (*Slice, 2002*). This method is the standard in GM and allows comparison of 2D or 3D configurations of landmarks within a common reference system (Fig. 3B), which is statistically estimated as the mean from the superimposition of the configurations after optimally minimizing the distances between homologous landmarks. This optimal superimposition is performed by translating, scaling and rotating the coordinates without altering the original distances between the landmarks (i.e., the topology of the configuration). The configurations are first brought to a common coordinate system that by consensus is the average configuration (or Grand Mean). The configurations are then rigidly scaled to the same size (i.e., isometric scaling), and they are subsequently rotated over the shared geodesic centroid. The residual mismatch and irreducible distance among homologous landmarks after the Procrustes alignment is due to the geometric differences between the configurations (after translation, rotation and scaling have been filtered out), and is known as Procrustes shape data; such data is suitable for further multivariate analyses. Importantly, these newly obtained data are 'invariant' to (i.e., it does not possess any information about) translation, scale and

rotation (i.e., posture) (see also; *Slice, 2007*; *Mitteroecker & Guntz, 2009*; *Viscosi & Cardini, 2011*; *Zelditch, Swiderski & Sheets, 2012*).

In most cases, the landmarks were digitized in lateral view in arbitrary orientations. However, in those specimens for which the LSC orientation within the skull was known, the landmark data were digitized with the LSC set at 0° (i.e., the LSC horizontal, see Table 1). Once the configurations have been aligned with GM any change in the orientation of the latter skulls corresponds to an angular change in the orientation of the LSC and can be measured. To be consistent with *Duijm*'s (*1951*) data, we maintained the author's notation of positive and negative values above and below the horizon, respectively. Importantly, the Procrustes data is invariant to translation, scale and rotation, and the way to depict the superimposed configurations is arbitrary. A logical orientation thus is often selected to neatly illustrate the results. Here we chose to orient the superimposed data according to the skull of *Crocodylus johnstoni*, digitized as if resting horizontally with its ventral surface on a flat surface (Fig. 2A) (*Witmer et al., 2008*). Given that the body plan of *Crocodylus johnstoni* is characteristically dorsoventrally compressed, there is little doubt about what is dorsal, ventral, anterior or posterior relative to Earth's spatial directions (i.e., the vector of gravity and the horizon). Thus, such selection guarantees that the dinosaur landmarks, once superimposed by the Procrustes method, will nearly share the same spatial coordinates relative to the Earth's axes. Furthermore, in such a posture the orientation of the LSC of *Crocodylus johnstoni* is nearly co-planar with the horizon (*Witmer et al., 2008*). Therefore, any angular differences in the semicircular canal's orientation that may result from the Procrustes superimposition (differences in the orientation of the LSC of each dinosaur taxon with respect to the horizontal one of *Crocodylus*) can be intuitively visualized. It is important to stress, however, that this way of depicting the results is as arbitrary as any other, it does not alter the results, and importantly, it does not have any functional meaning (i.e., postural).

In order to compare the geometric similarity between skulls we used a phenetic clustering algorithm (the unweighted pair group method, UPGMA (*Rohlf & Sokal, 1981*)) on the shape data, and compared this result with a UPGMA classification of the landmark configurations recovering rotational—postural—information (i.e., using the orientation of the LSC as a reference system in those specimens where this information was available).

## RESULTS

When the skulls are oriented relative to the Procrustes mean, the average orientation of the LSC with respect to the horizon is approximately −9° (Std. Dev. = 16.96°; Confidence interval ±95% = [11.67; 29.31]; Fig. 3), although this mean value may not be particularly informative given that deviations involve complementary (i.e., positive or negative) orientations (*Mardia, 1972*). Comparing the extremes (the prototyped skull of the rebbachisaurid *Nigersaurus* and the theropod *Incisivosaurus*), the total range of degrees of deviation relative to the horizon is approximately 55.7° (Range [Maximum = −51.1°, Minimum = 4.6]), which is nearly equivalent to the range of LSC orientations documented for living birds when their head is in the alert posture (approximately 50°; compare Figs. 2

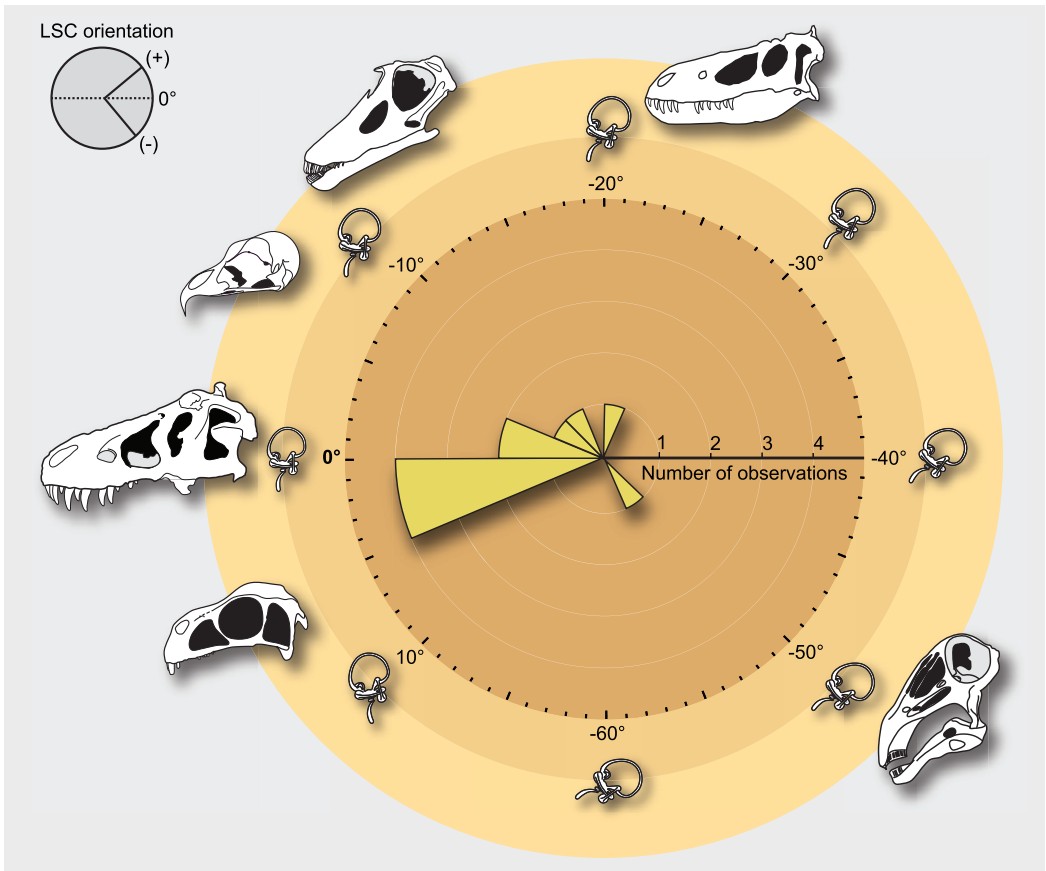

**Figure 4 Distribution of estimated measurements after Generalized Procrustes Analysis (GPA) of semicircular canal orientations for the studied dinosaur skulls.** In the distribution, the maximal range of angular variance spans between the skull of *Nigersaurus* (−51.1°) and that of *Incisivosaurus* (4.6°). The schematic skulls are shown in the posture obtained by the Procrustes alignment, and LSC orientations are measured relative to the horizon and as the difference between 0° and the new GPA orientation. Obtained LSC orientations for all dinosaurs after GPA are listed in Table 1.

and 4). The rebbachisaurid sauropod *Nigersaurus* is an outlier yielding a notable negative skewness to the distribution of LSC in saurischians. Although some skull postures after the Procrustes superposition do not differ much from the postures in which the LSC is horizontal, others differ more notably, and this is obviously due to the fluctuating orientations of their LSCs relative to their craniofacial geometry (Table 1). For instance, the skulls of *Tyrannosaurus* and *Majungasaurus* remain in a nearly identical position (i.e., its LSC remains nearly parallel—less than 1°—to the ground in both instances). In *Allosaurus* and *Camarasaurus* there is also a very slight deviation from their original posture (2.3° and −4.0°, respectively), whereas in the owl and in *Diplodocus* this deviation is definitely higher (−9.3° and −12.5°). In other skulls such as those of *Nanotyrannus* and *Nigersaurus* (both largely reconstructed fossils) the orientation of the canals changes significantly, (−22.3° and −51.1°, respectively). Although there is a tendency in the sampled dinosaurs to pitch up the LSC relative to the horizon, in *Incisivosaurus* the LSC is pitched down 4.6° (Table 1).

The phenetic clustering algorithm (UPGMA) on the Procrustes shape data finds two well defined groups differentiable on the basis of morphology (i.e., the orientation of the rostrum and the location of the nares) (Fig. 5, left column). All sauropods (including the prototyped skull of *Nigersaurus*) group together, and a parallel association happen with the skulls of the two tyrannosaurids in the sample (*Tyrannosaurus* and *Nanotyrannus*, the latter possibly a juvenile *T. rex*), which are also morphologically very similar to each other. However, all these congruent taxonomic groupings made by the UPGMA on the Procrustes shape data are dispersed if skulls are compared by re-setting the orientation of the landmarks so the LSC is at 0°; Fig. 5, right column).

## DISCUSSION

Orienting skulls for anatomical comparison using the LSC generally requires both the preservation of the canals, as well as CT scan data of sufficient quality to reliably delineate these structures. These criteria are not easily or often met (particularly for fossils), and thus finding skull orientations for comparative purposes using GM is more practical in the broadest range of cases, when general skull morphology can be restored. On the other hand, the obtained results stress that the orientation of the LSC of saurischian dinosaurs varies greatly relative to the rest of the skull (Fig. 4) and that such variability is independent of skull geometry (i.e., as in other tetrapods, including birds, there is no fixed alignment between the orientation of the LSC and skull morphology; *David et al., 2010*). When combined with information from living birds (*Duijm, 1951*), our results also show that a broad spectrum of LSC orientation remained relatively constant for at least the last 200 million years of dinosaur evolutionary history. In light of this, it is unlikely that the LSC serves as a consistent baseline to describe or compare craniofacial morphologies among these animals. Moreover, its use may lead to heterogeneous anatomical descriptions (Fig. 4), hence introducing inconsistencies when scoring character-states in cladistic analyses and inferring paleobiological attributes.

Before the establishment in GM of the Procrustes methods based on the Least Squares procedure, one way to register all landmark configurations to a common reference system (i.e., to standardize a set of coordinates for location, orientation, and size for comparative purposes) was to use the Two-point registration method (*Bookstein, 1991*), which establishes a fixed baseline between two landmarks. It was soon realized that truly invariant landmarks are extremely rare in complex forms, which entails that fixing any given two landmark locations to zero variance (i.e., as a baseline) inevitably and randomly transfers their true variation throughout the entire system (*Zelditch, Swiderski & Sheets, 2012*). This situation even worsens if the selected landmarks are too close, such as the two points that would define the plane of a LSC. A very similar situation takes place when establishing the LSC as a reference system. When a skull is oriented on the basis of aligning the labyrinth to a fixed set of coordinates (i.e., a horizontal LSC), the orientation of the LSC (with all its variability relative to other skull structures) is transferred to the orientation of the entire skull, resulting in an equivocal perception of the skull's geometry (i.e., confounding anatomical spatial directions across taxa). For example, using the

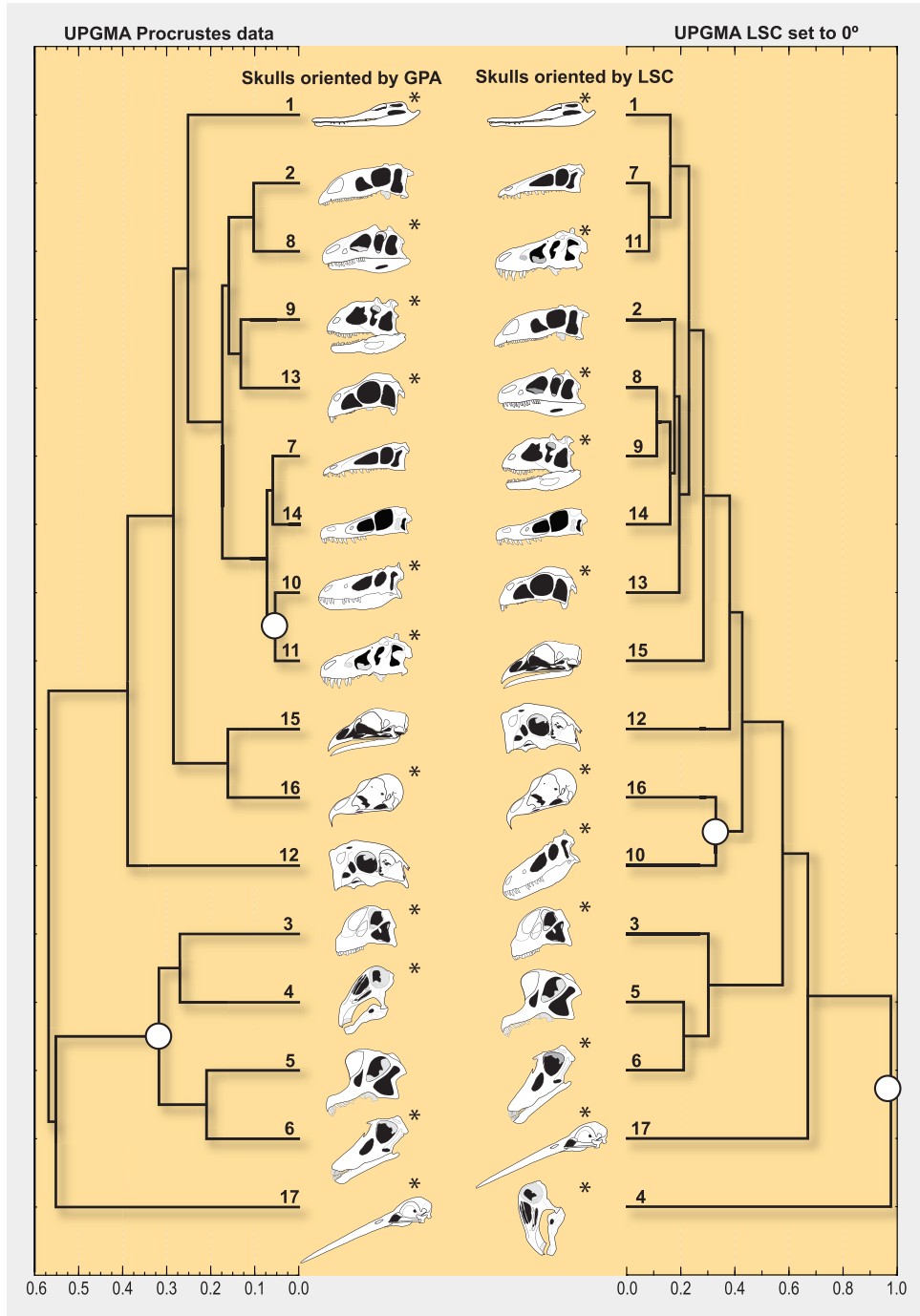

**Figure 5 UPGMA phenograms grouping dinosaur skulls by geometric similarity.** Separate columns illustrate the different skull postures obtained using Procrustes methods (left column) or by aligning the LSC with the horizon (right column). The branching diagram on the right column groups skulls comparing landmark data for which only translation and scale were filtered out, but not rotation (i.e., this data includes skull posture as morphological information, determined by the LSC set to 0°). The large white dots at the nodes are highlighting cases of notable grouping differences, such as considering the skull geometry of *Nigersaurus* as that of either a sauropod (left) 

**Figure 5 (...continued)**
or a bizarre dinosaur (right), and that of *Nanotyrannus* (presumably a juvenile *T. rex*) as different from that of *Tyrannosaurus*. The grouping in the right column indicates that rotation is a main source of morphological difference among skull geometries. The numbered terminal branches denote the taxa listed in Table 1; the LSC orientation is known for those with an asterisk.

LSC as a reference system, *Sereno et al. (2007)* envisioned the prototyped skull of the rebbachisaurid *Nigersaurus* as a very unusual animal because, among other morphological particularities, its rostrum pointed almost vertically towards the ground (Figs. 4 and 5). However, when the skull of *Nigersaurus* is oriented according to its craniofacial landmark homology with GM, its skull geometry is comparable to that of other sauropods (Fig. 5). Objectively, when viewed from the perspective of GM, it is the orientation of the labyrinth of *Nigersaurus* (not the head posture or its craniofacial geometry) that is unusual among other saurischians. Furthermore, given that the development of the semicircular canals has a strong genetic component (*Jeffery & Spoor, 2004*), it is plausible to assume that the orientation of the LSC is case-specific (*Billet et al., 2012*), although this needs to be further tested in extant taxa.

The use of the LSC as a reference system assumes that the orientation of the semicircular canal matches the coordinate system defined by Earth's gravity (*Vinchon et al., 2007*), presupposing that every species' stereotyped head posture (at rest or in alert) will be congruent with the vertical axis via the orientation of the LSC (*de Beer, 1947*). However, the fact that in every dinosaurian taxon the LSC does not share the same spatial orientation with respect to Procrustes-aligned craniofacial landmarks and to the horizon challenges that assumption. Such an assumption is also inconsistent with physiological evidence related to vestibular control, which indicates that LSC biomechanics sense angular acceleration stimuli and respond to head motion (*David et al., 2010*; *Fitzpatrick, Butler & Day, 2006*). In effect, there is a documented tendency in tetrapods to misalign the semicircular canals with the Earth's axes, which physiologically helps all canals to receive a component of angular acceleration during horizontal head rotations and thus, to actively participate in producing horizontal compensatory movements during motion (*Cohen & Raphan, 2004*). This vestibular control is jointly guided within the cerebellum through information provided by visual pathways specialized to detect translational visual flow (*Wylie, Bischof & Frost, 1998*; *Wylie & Frost, 1999*; *van der Water, 2012*). Furthermore, the characteristic physical organization of the labyrinth in three dimensions renders the canals biomechanically insensitive to the direction of gravity (*Rabbitt, Damiano & Grant, 2004*) and that function is restricted to the otolith within the vestibular system.

In most anatomical descriptions of archosaur skulls, the plane of reference used to align skulls for comparison is not specified. Judging by figures in many of these studies, this is most typically done by orienting the maxillary tooth row horizontally (e.g., *Sampson et al., 2010*; *Horner & Goodwin, 2009*; *Campione & Evans, 2011*). In order to be consistent in anatomical descriptions, here we recommend that, (1) the frame of reference for aligning anatomical axes should be stated, and (2) that the axes obtained using the geometric morphometric methods described here are easily standardized and

reproducible, either in 2D or 3D. Whereas our results do not address the possibility of estimating how fossil organisms held their heads (i.e., how the head was carried), an issue that needs to be further studied in living organisms, they clearly show that the horizontal alignment of the LSC cannot provide a consistent anatomical reference system for skull comparisons in dinosaurs, and possibly in other tetrapods. Instead, we argue that by analyzing homologous morphological landmarks, Geometric Morphometrics offers a more consistent anatomical reference system, one that is independent of posture and purely based on homologous anatomical variables.

## ACKNOWLEDGEMENTS

We thank C Mehling (American Museum of Natural History, New York) and D Unwin (Museum für Naturkünde, Berlin) for specimen access and A Balanoff for the CT scans of *Incisivosaurus*. We are also grateful to S Abramowicz and W Evans for creating the artwork and editing the manuscript, respectively. J Lobón-Cerviá, A Buscalioni, and N Martínez-Abadías provided helpful comments to the manuscript and L Witmer and M D'Emic reviewed an earlier version of the paper—all these discussions strengthened our study.

### Funding

Funding for this research was provided by a Collection Study Grants of the American Museum of Natural History (New York), the MECD/Fulbright Posdoctoral Mobility Program (Spain), project DGCYT CGL2009_11838 BTE from the Ministerio de Economia y Competitividad (Spain), and the Dinosaur Institute of the Natural History Museum of Los Angeles County (Los Angeles). The funders had no role in study design, data collection and analysis, decision to publish, or preparation of the manuscript.

### Grant Disclosures

The following grant information was disclosed by the authors:
Ministerio de Economia y Competitividad (Spain): DGCYT CGL2009_11838 BTE.

### Competing Interests

Jesús Marugán-Lobón, Luis M. Chiappe and Andrew A. Farke are Academic Editors for PeerJ. There are no other competing interests.

### Author Contributions

- Jesús Marugán-Lobón conceived and designed the experiments, performed the experiments, analyzed the data, contributed reagents/materials/analysis tools, wrote the paper, general discussion.
- Luis M. Chiappe conceived and designed the experiments, performed the experiments, analyzed the data, contributed reagents/materials/analysis tools, wrote the paper, general discussion.
- Andrew A. Farke wrote the paper, general discussion.

## Supplemental Information

Supplemental information for this article can be found online at http://dx.doi.org/10.7717/peerj.124.

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
