# Peer review of "The variability of inner ear orientation in saurischian dinosaurs: testing the use of semicircular canals as a reference system for comparative anatomy"

_PeerJ, doi:10.7717/peerj.124_

## Round 0.1 · original submission · Minor Revisions

The reviewers are in firm agreement that the study is publishable and I concur that just minor revisions are needed. In particular, a better treatment of how head posture was assessed, especially in animals where it was not known, needs to be given and more consideration of the assumptions made in assignments of head posture. Reviewer 2 makes the excellent point that head posture in extant archosaurs is generally unknown (to a rigorous degree- if applying rigorous GMMs you should be equally concerned about rigorous measurements of actual head posture since this is the key question!) so some qualifiers are needed that this is not a definitive analysis (but nor has any study been that!). Otherwise the corrections are quite small. Nice study and thanks for sending it to PeerJ with open review!

Reviewer 1 ·

Basic reporting

This study is highly relevant for publication in PeerJ and is likely to be of wide interest to palaeobiologists and anatomists. As far as I can see it does not contravene PeerJ policies. I assume that all the skull outline images and the bird skull in figure 2 are either produced by the authors and/or do not contravene any copyright rules. Perhaps the authors could confirm this.

Experimental design

The authors test the assertion that orientating the lateral semicircular canal (LSC) of the inner ear to the horizontal serves as a proxy for estimating skull posture. They use geometric morphometrics (GMM) to ascertain any discrepancy between LSC derived skull orientations and GMM derived skull orientations.

I'm generally happy with the experimental design with the exception of a couple of comments.

Firstly, this paper would benefit from a more in depth introduction to Duijm's 1951 paper, which the majority of the readership will not be familiar with and will not be able to access. For example, how did Duijm record the orientation of the SSCs? By dissection of the species he/she observed? Also in the methods section, the authors need to make it clear that they have plotted/replotted Duijm's data as a first step in their analysis. There is nothing wrong with this, as they've resurrected an interesting dataset that is pertinent to the question the authors seek to address here, but I think that this exercise should be discussed in the methods.

I would like the authors to provide more detail about orientating the skulls in Table 1. Out of the 17 taxa listed here, 8 of the 13 dinosaurs have data on LSC orientation, as does the Johnstons crocodile, and two of the three birds. The authors state on line 131 that "each skull posture was digitized with the LSC set at 0 degres, [sic] in those specimens for whom this information was available". But how did the authors treat those skulls for which LSC was not available?

Validity of the findings

The data is presented in an easily visualised and understandable manner. I have no queries about this part of the paper. I have examined and read through the tps and protocol files in the Supp Info and this represent an excellent step by step guide to recreating skull posture in additional specimens.

Additional comments

Typos and suggested edits:

- In the abstrast, suggest "If such were the case..." to be replaced with "If this were the case..."
- The final sentence of the abstract- "geometric morphometrics provides a more consistent method for establishing comparisons among dinosaur taxa" - can you be more specific about what you mean re: anatomical comparisons. Are you refering to comparisons of posture?
- lines 52, 53 - "skul modifications similar to primates" - explain what you mean here. It isn't immediately obvious what you are refering to.
- Line 68 - it would be useful if the authors could explain further what they mean by 'morphological comparisons'. The term sounds vague.

·

Basic reporting

These comments relate to very minor changes, otherwise the MS would be acceptable as is.

1. “Procrustes methods” are usually just referred to as ‘geometric morphometrics’ in studies that use the approach. Much as I like Greek mythology it might be better to stick to ‘geometric morphometrics’ throughout, especially as it’s explained (ln. 79) that Procrustean shape analysis is normally just referred to as geometric morphometrics. Anyone reading this will be more familiar with the latter term.

2. There are a number of instances where the word ‘data’ is treated as singular (e.g., ln. 128 “this newly obtained data”). These instances need to be corrected.

3. Another plural problem – ln. 150: “each dinosaur taxa”. This needs to be checked and corrected throughout.

4. Ln. 188: Should the statement about _Nannotyrannus_ being a possible juvenile _T. rex_ have a citation?

Experimental design

No Comments

Validity of the findings

No Comments

Additional comments

This is a rather neat study that tests whether the lateral semicircular canal (LSC) is actually useful as a horizon index plane for determining head carriage, and for relating craniofacial morphology variation to a common datum. Such a test is particularly important to carry out in view of the rise in X-ray CT analysis of fossils in recent years; the ease in which orientation of the LSC can now be determined in fossil skulls has led to the LSC routinely being trouped out to demonstrate ‘alert’ head carriage in extinct taxa. As you note, the use of the LSC has been questioned, although Larry Witmer did make a reasonable defence of its use in my view. However, your study provides convincing evidence that the situation is not clear cut, and the MS is potentially a very important contribution to vertebrate palaeontology.

As someone who has also often used LSC orientation as a proxy for head carriage in birds, I am convinced by this study that geometric morphometrics (GM) is probably a better approach. In what I am about to say I want to make clear that I do not believe the work should be extended or altered for this publication - that is why I have entered 'No Comments' in the Experimental design box. However, I can see a few criticisms that are likely to be used to argue against your conclusions, and I would suggest adding some text to counter these in advance.

One problem is that head carriage in extinct saurischian dinosaurs is essentially an unknown, so it could be argued that the relationship between the landmark data, actual ‘alert’ head position and LSC orientation must first be ‘ground truthed’ by analysing a reasonable selection of extant archosauromorphs (birds and crocs) for which ‘alert’ head position is actually known. There are arguably not enough extant taxa in the dataset to do this. You rightly note that many avian taxa have huge variations in skull structure (ln. 52), but though the three taxa included are very different from each other, they don’t encompass enough of the variation seen in living clades. For instance, the skull axis flexion in parrots is very different to sandpipers, cormorants or flamingos (or for that matter, owls in this dataset). This could be countered if the study is presented more as a proof-of-concept that shows the uses of GM for assessing craniofacial variation, and demonstrates rigorously the problems with using the LSC alone for determining alert head posture.

I may have misunderstood this in the methods section, but another criticism might be that the study appears to apply a 2D superposed landmark approach to images of skulls in lateral view. Error can be introduced in landmark placement due to foreshortening in a 2D image; this has been a problem with GM analyses of skulls in dorsal view and it has been discussed in the literature. However, this would not be a problem if 3D landmarks were placed on digital 3D models. If this is the case it may need to be made clearer, if not a rationale for why error should not affect results needs to be given (e.g., use with crushed bird fossils). Of course, I do realise the main thrust of the study concerns dinosaurs rather than birds.

Leading on from above, another problem is that the technique will be problematic for a number of fossil taxa, and especially birds – many three dimensionally-preserved birds lack a rostrum, or the rostrum is incomplete (from my own work, the rostrum in the holotype of _Odontopteryx_ is incomplete, and it’s missing entirely in _Halcyornis_, _Cerebavis_ and the 3D specimen of _Patagopteryx_). While the statement about GM being more useful in a broad range of cases (ln. 198) is true, it should probably be emphasised that the methods used in this study are well suited to roadkills in which vestibular reconstruction is impossible.

Some minor comments to consider:

1. Ln. 81: It’s not strictly true that all GM approaches rely on homology for landmark selection. Eigensurface takes into account whole object shape, and apart from the start and stop points, this is basically true for standard eigenshape and similar techniques.

2. Ln. 235: it may be worth citing Billet et al. (2012) as a case where labyrinth morphology is affected by weakening of function-related drivers (Billet, G, Hautier, L, Asher, R, Schwarz, C, Crumpton, N, Martin, T, Ruf, I, (2012). High morphological variation of vestibular system accompanies slow and infrequent locomotion in three-toed sloths. Proceedings of the Royal Society B: Biological Sciences, 279, 3932–3939).

3. Lns 245-9: This is true, but probably not exactly for the reasons stated – see Wylie and Frost (1999) for a discussion of azimuth tuning in the vestibulocerebellum of pigeons (Wylie, RWW and Frost, BJ (1999). Complex Spike Activity of Purkinje Cells in the ventral uvula and nodulus of pigeons in response to translational optic flow. Journal of neurophysiology, 81(1):256-66).

Stig Walsh

---

## Round 0.2 · accepted · Accept

I am very satisfied with the revisions, so congratulations- your paper is accepted! Peer review worked quite well here.

A very minor note-- the Response document included the authors' tracked changes from prior versions of that document. Be watchful with that in the future- accept all changes in the document before submitting to the journal! :)